# The Curing Kinetics of E-Glass Fiber/Epoxy Resin Prepreg and the Bending Properties of Its Products

**DOI:** 10.3390/ma14164673

**Published:** 2021-08-19

**Authors:** Lvtao Zhu, Zhenxing Wang, Mahfuz Bin Rahman, Wei Shen, Chengyan Zhu

**Affiliations:** 1College of Textile Science and Engineering (International Institute of Silk), Zhejiang Sci-Tech University, Hangzhou 310018, China; zhenxing@zstu.email.cn (Z.W.); mahfuz.zstu@gmail.com (M.B.R.); cyzhu@zstu.edu.cn (C.Z.); 2Shaoxing Baojing Composite Materials Co., Ltd., Shaoxing 312000, China; shenw@jinggonggroup.com

**Keywords:** prepreg, glass fiber/epoxy resin, curing kinetics, bending properties

## Abstract

The curing kinetics can influence the final macroscopic properties, particularly the three-point bending of the fiber-reinforced composite materials. In this research, the curing kinetics of commercially available glass fiber/epoxy resin prepregs were studied by non-isothermal differential scanning calorimetry (DSC). The curing kinetic parameters were obtained by fitting and the apparent activation energy E_a_ of the prepreg, the pre-exponent factor, and the reaction order value obtained. A phenomenological *n*th-order curing reaction kinetic model was established according to Kissinger equation and Crane equation. Furthermore, the optimal curing temperature of the prepregs was obtained by the T-β extrapolation method. A vacuum hot pressing technique was applied to prepare composite laminates. The pre-curing, curing, and post-curing temperatures were 116, 130, and 153 °C respectively. In addition, three-point bending was used to test the specimens’ fracture behavior, and the surface morphology was analyzed. The results show that the differences in the mechanical properties of the samples are relatively small, indicating that the process settings are reasonable.

## 1. Introduction

Glass fiber reinforced composites have remained interesting for academia as well as industry due to their outstanding properties, i.e., lightweight, fatigue resistance, corrosion resistance, high specific strength, and easy manufacturing techniques, as well as reasonable cost compared to other fiber-reinforced composites [1,2]. They have been widely used in many areas such as aerospace, marine, automotive, medical, sports, machine tools, and for several other structures [3]. E-glass fiber/epoxy resin prepreg is an intermediate for the manufacture of composite materials [4]. It is a composition obtained by impregnating E-glass fiber or fabric with epoxy resin. The epoxy resin in a B-level semi-cured state can be cured under a specific temperature and pressure to obtain composite products. This manufacturing method can significantly improve production efficiency but has high requirements for the rationality of the curing process design [5]. This is mainly reflected in the setting of process parameters. Setting reasonable process parameters is the key to obtaining high-quality products.

The study of curing kinetics is an important basis for designing curing conditions [6,7]. At present, the curing kinetics of resin matrix composites are mainly based on the mechanism method and phenomenological method [8,9]. The mechanism method regards the overall reaction as the sum of each elementary reaction and studies each elementary reaction and the related effects. This can accurately describe the curing kinetics, but it is difficult to calculate and model [10]. The phenomenological method is semi-empirical curing, i.e., the kinetic model. It mainly studies the overall reaction and obtains the kinetic parameters through a mathematical simulation from the empirical equation. It has the advantage of being simple and intuitive, so it is widely used. Cai et al. used differential scanning calorimetry under dynamic as well as isothermal conditions to study the curing kinetics of two epoxy resin/flexible amine systems and established a phenomenological curing kinetic model. The experimental results have consistency with the predicted results of the model; thus, these models can be used to describe the curing phenomenon of the resin well [11]. Liu et al. investigated the phenomenological curing model of epoxy resin for prediction of the degree of the cure based on an autocatalytic cure kinetic model and a new modified phenomenological curing model based on Olivier’s model, and they showed that the modified Olivier’s model is simple, easy to use and more accurate [12]. The curing kinetics has a significant influence on the macroscopic mechanical properties of composite materials [13].

The three-point bending experiment is considered as one of the important and most extensively used tests in measuring the mechanical properties of composite material [14]. Mouzakis et al. used a three-point bending test to study the changes in the bending strength of polyester/glass fiber reinforced composite materials (GFRPC) under the combined effects of temperature, humidity, and ultraviolet radiation [15]. Ren et al. prepared a modified fiber-reinforced phthalonitrile-based resin composite laminate and tested its bending strength through a three-point bending test [16]. Jariwala et al. studied the influence of fiber structure on the mechanical properties of E glass fiber reinforced composites, and the bending strength was tested by a three-point bending experiment [17]. Shin et al. studied the optimal conditions of glass fiber/epoxy resin composites based on the mixing ratio of the two epoxy resin matrices. Three-point bending was used to test the bending performance and finally obtained when the epoxy resin mixing ratio is 1:1, the mechanical performance of the composite material is the best [18]. Therefore, we choose a three-point bending experiment to test the sample’s mechanical properties in our research.

As the curing process parameters have a crucial influence on the quality of composite products, we focused on the curing kinetics of epoxy resin. The curing kinetic parameters were obtained by non-isothermal differential scanning calorimetry, the curing kinetic model of the resin was established, and the optimal curing temperature parameters of the resin were obtained by the T-β extrapolation method. Further, the composite laminates were prepared by the vacuum hot pressing process, and three-point bending was used to test the difference in mechanical properties between different samples. This method is of great significance to guide the actual production; it can quickly obtain the best curing temperature range of the resin and quickly check its quality.

## 2. Materials and Methods

### 2.1. Materials

The sample used was a prepreg (composed of ER468-2400 E-glass fiber and 7901 DNK toughened epoxy resin) purchased from Weihai Guangwei Composite Material Co., Ltd., Weihai, China. The 7901 DNK is a resin with good curing and process properties obtained after modification by a variety of epoxy resins, mainly including DGEBA and BDGE. IPDA is the main curing agent for this resin. Their structure are shown in the Figure 1. The main properties of prepreg are shown in Table 1.

### 2.2. Manufacturing of Composite Laminates

The prepregs were cut to 250 × 300 mm and laid up to 15 mm on the mold by hand, laying 5 groups of the same thickness to detect the difference in mechanical properties after curing. After being laid up, vacuum treatment was used to eliminate minute bubbles that may occur during the layup process so that the good mechanical properties of the laminate could be obtained. Finally, the laminates were put into a vacuum autoclave for curing using 1.33 MPa pressure. The curing temperature curve of the prepreg was set according to the DSC test results.

### 2.3. Three-Point Bending Test

A sample of 270 × 13 × 15 mm was cut out from the middle of each group of laminates, and a total of five samples were obtained. The samples were placed on a universal testing machine (M.T.S. landmark 370.10, Eden Prairie, MN, USA) to test the three-point bending performance. The span was set to 240 mm (shown in Figure 2), and the upper indenter remained stationary. The lower indenter is loaded upwards at a speed of 10 mm/min so that the sample is stressed at three points, and finally, a fracture occurs in the middle of the sample. The test was carried out under the conditions of a temperature of 25 °C and a humidity of 50%.

### 2.4. Characterization

DSC 25 (T.A. Instruments, New Castle, DE, USA) was used for calorimetric measurements for dynamic DSC scans. Three samples were heated from 25 to 250 °C at different heating rates, i.e., 5, 10, and 15 °C/min, with approximate sample weights of 3.50 mg under an N_2_ purge at 50 mL/min. The DSC was used to determine the heat released while curing the specimen under non-isothermal conditions. The thermal weight loss of the sample was determined using PerkinElmer PYRIS 1 thermogravimetric analysis (TGA). Then, 5–8 mg of the sample was heated from 25 to 1000 °C at a rate of 10 °C/min under a nitrogen purge of 30 mL/min. A scanning electron microscope (SEM, COXEM, Daejeon, Korea) operating at 10 kV was used to observe the SEM images surface morphology of the fractured surface of the composite laminates.

## 3. Results and Discussion

### 3.1. Thermogravimetric Analysis

The TGA curve of the epoxy resin sample is shown in Figure 3. The weight loss process of the sample is roughly divided into three processes: the first stage is before 350 °C, the weight loss rate is about 5%; the second stage (350–550 °C) has the fastest weight loss, which is about 85% lower; in the third stage (550–1000 °C), the mass loss of the sample tends to be stable, with a loss of about 2.88%. The initial weight loss (≤5%) was attributed to the removal of water molecules and some small organic compounds from the resin [19]. The temperature at 5% weight loss (*T_d_*_5_) was 350 °C. The thermal decomposition temperature at 30% weight loss (*T_d_*_30_) was 405 °C, far more significant than any service temperature proposed for this resin. The peak degradation temperature (T_deg_) was determined by differentiating the degradation curve, as shown in Figure 3. T_deg_ exhibited a single-step degradation profile which suggests that the sharp decomposition of the crosslinked polymer network. The heating program was designed to obtain the thermal degradation temperature of the epoxy resin sample to ensure that the sample will not be thermally degraded during the DSC test, so the maximum temperature of the DSC test was set to 250 °C.

### 3.2. DSC Curve Analysis

The DSC test results of the prepreg under different heating rates (*β*) are shown in Figure 4. The DSC curves of the prepreg under different heating rates all showed a single exothermic peak, which indicated that the curing reaction was completed in one step. This is because the heat flow rate is directly proportional to the heating rate. As the heating rate increases, the start temperature (*T_i_*), peak temperature (*T_p_*), and end temperature (*T_f_*) of the curing reaction also increase. As the heating rate increases, the heating effect per unit time increases, and the thermal inertia and temperature difference become larger [20]. 

The curing process of thermosetting resin is complex, involves a series of chemical reactions, and finally transforms the low molecular weight monomer into the macromolecular crosslinked network [13]. Under normal circumstances, this process is accompanied by a change in heat. The rate of heat generated was plotted against time, and their integration provided the amount of heat released. According to the curve shown in Figure 4, total enthalpy was calculated by the area under the exothermic curve of the DSC. This is an essential physical quantity representing the total heat released during the curing reaction, as shown in Table 2—the amount of heat released by the curing reaction increases as the heating rate increases.

The curing of the epoxy resin is a staged process; there are usually three stages of curing, commonly referred to as pre-cure, cure, and post-cure, and their respective temperatures obtained on the DSC curves are termed as initial temperature (*T_i_*), peak temperature (*T_p_*), and final temperature (*T_f_*) [21], as shown in Figure 5; the temperature at which the curing reaction occurs is linearly related to the heating rate. So, the curing process temperature is usually determined by extrapolation. According to the T-β extrapolated curve, when β is zero, the theoretical gel temperature ($T_{i}^{*}$), theoretical curing temperature ($T_{p}^{*}$), and theoretical post-treatment temperature ($T_{f}^{*}$) of the prepreg is 116, 130, and 153 °C, respectively. That is to say; the theoretically reasonable curing process is to heat the prepreg to 116 °C. The viscosity of the resin will gradually decrease during this heating process, which is conducive to the resin infiltration of the reinforcing fibers; then the temperature is increased to 130 °C, the resin is crosslinking, and curing occurs at this temperature; finally, keep it at 153 °C for a period of time to completely cure the resin.

### 3.3. Curing Kinetics

The epoxy cure kinetics are well represented by the phenomenological models. It is intuitive and straightforward, does not require a large amount of experimental data. The “degree of cure” is the measure of the chemical reactions (bond exchanges) that occurred during the curing process of the resin, and often it is the true representative of crosslinking density or molecular weight [22,23]. The degree of cure, α, is determined by Equation (1) [24]:(1)$$\alpha =\frac{H(t)}{H_{tot}}$$
where *H*(*t*) is the heat of reaction at time *t*, and *H_tot_* is the total heat of the reaction. The relationship between the degree of curing of the resin sample and the temperature is shown in Figure 6. As the heating rate increases, the temperature required for the resin to reach the same degree of curing continues to increase. This is mainly because the reaction rate increases with the increase of the heating rate, resulting in a faster increase in the system’s viscosity. Part of the monomers is too late to react or cannot diffuse into the gel system in time, and the resulting temperature gradient increases, resulting in a decrease in the curing of the system during the diffusion phase at the same temperature [20].

The rate of the kinetics process in kinetic analysis can be described by Equation (2) [25]: (2)$$\frac{d\alpha }{dt}=K(T)f(\alpha )$$
where *K*(*T*) is the reaction rate constant (temperature-dependent), and *f*(*α*) describes the phenomenological reaction model. *K*(*T*) can be mathematically presented according to the Arrhenius relationship.
(3)$$K(T)=A\exp(-\frac{E_{a}}{RT})$$
where *A* is the pre-exponential factor, *E_a_* is the activation energy, and *R* is the universal gas constant.

Comment model for *f*(*α*) is the *n*th-order model, provided in Equation (4) [26]:(4)$$f(\alpha )={\left(1-\alpha \right)}^{n}$$

Combination of Equations (2)–(4) results in the final kinetic expression provided in Equation (5):(5)$$\frac{d\alpha }{dt}=A\exp(\frac{-E_{a}}{RT})\cdot {\left(1-\alpha \right)}^{n}$$

During the dynamic heating process of prepreg, the curing rate is a function of temperature and curing degree. A common technique for evaluating these data is the Kissinger method, which can be used to determine the activation energy of the curing reaction [27,28], as presented in Equation (6):(6)$$\ln(\frac{\beta }{T_{p}^{2}})=\ln(\frac{AR}{E_{a}})-\frac{E_{a}}{RT_{p}}$$

The fitting the linear relationship between $\ln\left(\frac{\beta }{T_{p}^{2}}\right)$ versus $\frac{1}{T_{p}}$ can provide value of apparent activation energy *E_a_*, the pre-exponential factor A through the intercept. The reaction order *n* can be calculated according to the Crane Equation [29,30].
(7)$$\frac{d(\ln\beta )}{d(\frac{1}{T_{p}})}=-\frac{E_{a}}{nR}$$

According to the data in Table 3, using $\ln\left(\frac{\beta }{T_{p}^{2}}\right)$ and $\frac{1}{T_{p}}$ to plot, and then perform linear fitting, the obtained epoxy resin curing reaction activation energy and pre-finger factor fitting curve is shown in Figure 7a. The slope of the fitted curve is −8.41, and the intercept is 9.95. The Adj. R-square is 0.9933, indicating a good linear correlation. Substituting the obtained slope and intercept into Equation 6 can be calculated: *E_a_* = 69.95 kJ/mol, A = 1.76 × 10^5^ min^−1^.

Use lnβ and $\frac{1}{T_{p}}$ to the plot, and then perform the linear fitting. The reaction order fitting curve of the epoxy resin curing reaction is shown in Figure 7b. The slope of the fitted curve is −9.26, the intercept is 24.04, and the Adj. R-square is 0.9944, indicating a good linear correlation. Substituting *E_a_* and the slope and intercept of this curve into Equation (7) can find *n* = 0.91.

Substituting the obtained curing kinetic parameters *E_a_*, *A*, *n* into the *n*th-order curing kinetic Equation (5) and integrating them, the following Equation (8) can be obtained, reflecting the relationship between the curing degree and the reaction temperature and time.
(8)$$\alpha (t)=1-{\left[1-1.58\times 10^{4}\exp\left(-\frac{8.41\times 10^{3}}{T}\right)t\right]}^{11.11}$$

Compared with other binders, this article uses a modified hybrid epoxy resin whose main components include aliphatic and cycloaliphatic resins. Similar research comparisons with other systems are shown in Table 4. It is found that both the apparent activation energy *E_a_* and the reaction order *n* are relatively small. In the resin system of this article, the apparent activation energy is relatively low. This may be because as the degree of curing increases, the free movement between molecular segments is less restricted, and the degree of freedom is higher, so more energy is not needed to overcome the energy barrier. The concentration of the resin system has a small effect on the reaction rate, so the reaction order is small.

### 3.4. Bending Performance

The load-displacement curves and three-point bending test data of the five samples are shown in Figure 8 and Table 5. The stress–strain curves of the samples increase linearly at the initial stage; after reaching the maximum strain, the bending stress suddenly drops, showing prominent brittle fracture characteristics. It is worth noting that before the maximum stress is reached, there is no obvious inflection point in the curve, indicating that the sample has not been significantly damaged before the final failure. 

Figure 9 shows the surface morphology of the sample after being destroyed. The damage of the specimen mainly occurs at the compressed place, as shown in Figure 9a, because the compressed place has the most significant displacement. When the load is generated, the epoxy resin is first subjected to the force, and then the force is transmitted to the glass fiber that plays the main force-bearing role. As the load increases, downward strain occurs at the compressed area, resulting in mutual extrusion. Epoxy resin has a poor ability to withstand force. It is first damaged and then crushed, leaving only a small part on the glass fiber, as shown in Figure 10a. When the stress reaches the maximum that the glass fiber can withstand, the glass fiber breaks; one end of the broken fiber continues to move downward under the squeeze of the upper indenter, pulling the fiber along the direction of the yarn, causing cracks to occur and expand. The cracks propagate to form different fiber bundles, as shown in Figure 10b. The failure mode of the sample is mainly manifested as the slippage and mutual extrusion of the fiber bundles, which is easier to be observed in the macroscopic view, as shown in Figure 9b. This is because the resin is tightly combined with the fiber and is not easily split into individual fibers.

## 4. Conclusions

Herein, the cure kinetics of an E-glass/epoxy resin prepreg was investigated by DSC. The fracture behavior of the composite was studied by a three-point bending test, and surface morphology was assessed. The TGA analysis showed a stable thermal structure; the resin decomposes rapidly at about 350 °C. DSC studies showed that the optimal pre-curing temperature, gel temperature, and post-treatment temperature of prepreg are 116, 130, and 153 °C, respectively. Based on the Kissinger and Crane Equations, the apparent activation energy *E_a_* of the prepreg is 69.95 kJ/mol, the pre-exponent factor A is 1.76 × 10^5^ min^−1^, and the reaction order *n* is 0.91. Finally, the *n*th-order curing reaction kinetic equation of the prepreg is established. A three-point bending test was performed on the laminate sample. The results show that the difference in mechanical properties is relatively small, indicating that the laminate has fewer internal defects, stable quality, and reasonable curing temperature process design. The SEM photos also prove that the resin and fiber are tightly bonded.

## Figures and Tables

**Figure 1 materials-14-04673-f001:**
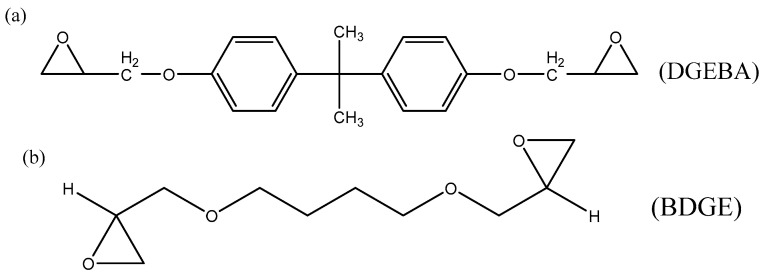
Chemical structure formula of (**a**) DGEBA, (**b**) BDGE and (**c**) IPDA.

**Figure 2 materials-14-04673-f002:**
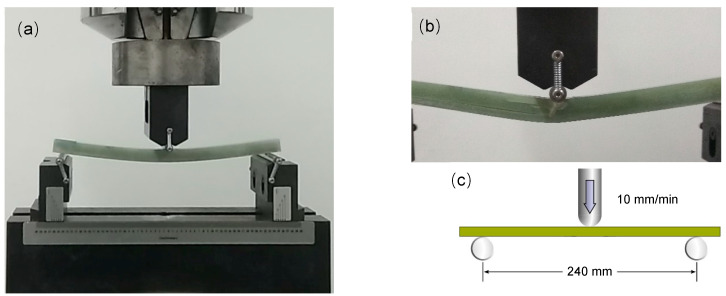
Three-point bending test diagrams: (**a**) specimen under bending load; (**b**) specimen fracture closer view; (**c**) schematic diagram.

**Figure 3 materials-14-04673-f003:**
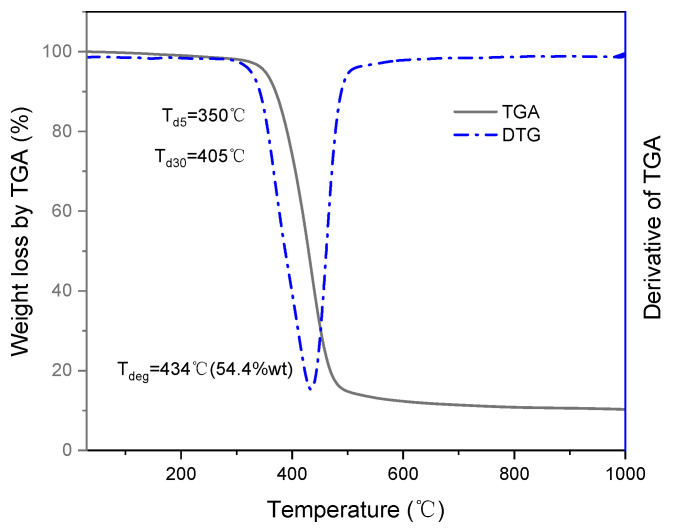
TGA graph of 7901 DNK toughened epoxy resin.

**Figure 4 materials-14-04673-f004:**
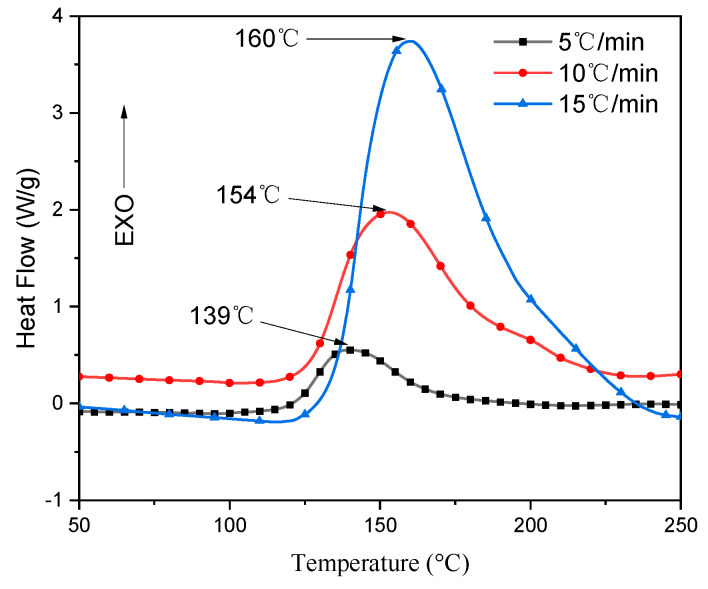
DSC curve for the 7901 DNK toughened epoxy resin at different heating rates of 5 °C/min, 10 °C/min, 15 °C/min in N_2_ atmosphere.

**Figure 5 materials-14-04673-f005:**
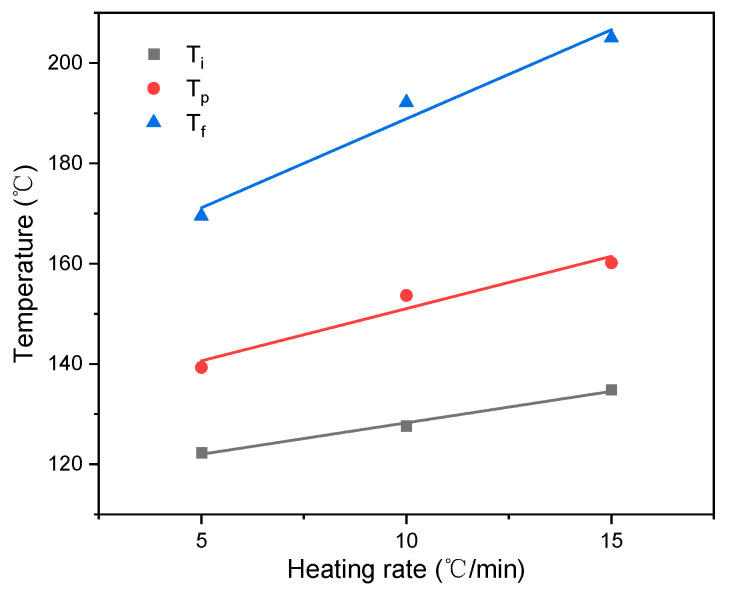
Extrapolation curves of the curing temperature.

**Figure 6 materials-14-04673-f006:**
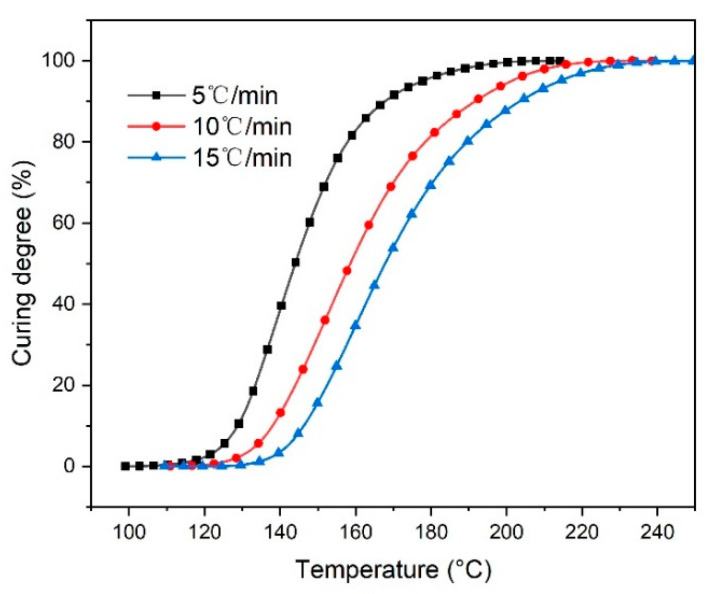
Curves of the curing degree temperature.

**Figure 7 materials-14-04673-f007:**
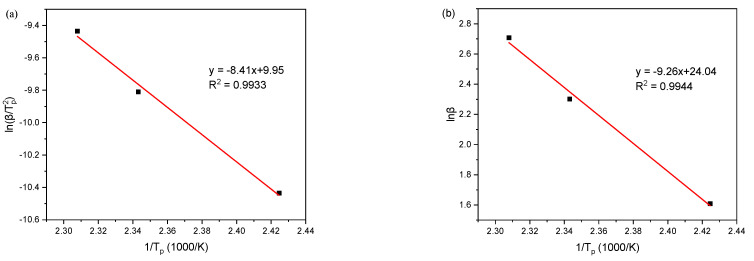
The fitting curve of (**a**) activation energy and pre-exponential factor; (**b**) reaction order.

**Figure 8 materials-14-04673-f008:**
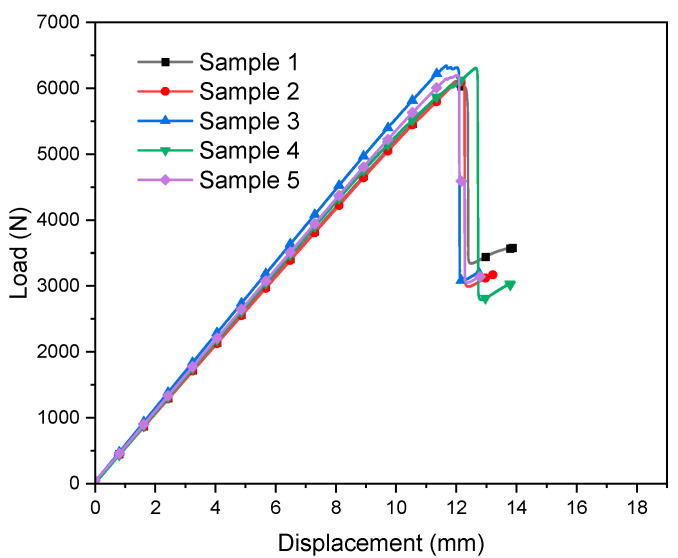
Load–displacement curve of E-glass fiber/epoxy composite samples.

**Figure 9 materials-14-04673-f009:**
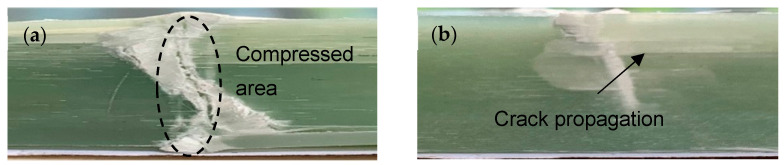
SEM images of fracture surface with (**a**) 500× magnification and (**b**) 80× magnification.

**Figure 10 materials-14-04673-f010:**
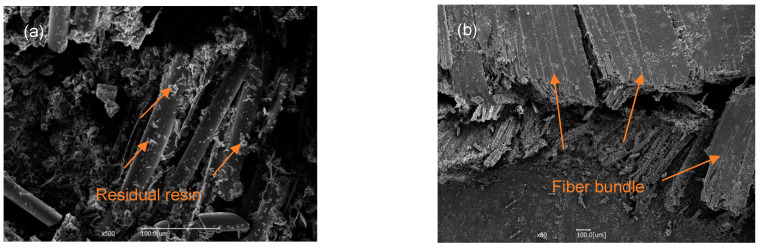
SEM photos of the fracture surface with (**a**) 500× magnification and (**b**) 80× magnification.

**Table 1 materials-14-04673-t001:** Main properties of prepreg.

Physical Properties	Unit	Value
The weight of prepreg per unit area	g/m^2^	2352 ± 5%
Fiber weight per unit area	g/m^2^	1600 ± 5%
Single-layer thickness	mm	1.2 ± 0.1
Resin content	%	32.0 ± 3.0
Volatile content	%	≤1.0

**Table 2 materials-14-04673-t002:** Thermodynamic parameters of prepreg at different heating rates.

*β*/(°C·min^−1^)	Cure Temperature (°C)			*H_tot_*/(J·g^−1^)
	*T_i_*/°C	*T_p_*/°C	*T_f_*/°C	
5	122	139	169	257
10	128	154	192	499
15	135	160	205	780

**Table 3 materials-14-04673-t003:** DSC exotherm peak temperatures of resin at different heating rates.

β/(°C∙min^−1^)	*T_p_*/K	1/*T_p_*(1000/K)	lnβ	$ln\left(\beta /T_{P}^{2}\right)$
5	412	2.42	1.61	−10.43
10	427	2.34	2.30	−9.81
15	433	2.31	2.71	−9.43

**Table 4 materials-14-04673-t004:** Comparison with similar studies of other systems.

Type of Bender	*E_a_* (kJ/mol)	Reaction Order	Ref Number
Bisphenol-A	71.42	1.3435	[31]
TDE-85/BMAPF	73.99	0.908	[32]
Multifunctional P-containing	72.96	3.21	[33]
Phenolic	96.03	-	[34]
VTM264	96.4	-	[35]
E51	102.55	1.3064	[36]
Epoxy	88.549	1.323	[37]
7901 DNK	69.95	0.91	This work

**Table 5 materials-14-04673-t005:** Three-point bending test data.

Sample	Load Time (s)	Displacement (mm)	Load (N)	Flexural Modulus (MPa)
1	96.93	16.82	6112	411
2	75.72	13.22	6106	408
3	72.95	12.76	6356	438
4	79.21	13.79	6315	421
5	74.34	12.99	6231	423

## Data Availability

The data can be provided by the corresponding author on request.

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
