# Peer review of "The Curing Kinetics of E-Glass Fiber/Epoxy Resin Prepreg and the Bending Properties of Its Products"

_materials, 2021, doi:10.3390/ma14164673_

Round 1
Reviewer 1 Report
In my opinion, the work is quite interesting, albeit small.
I believe that the work should be supplemented with experimental data, for example, to determine the thermal characteristics for other systems (with a changed ratio, the binder is a reinforcing filler, another binder, etc.). At the moment, the work seems to be very small (all the work can be done in a week).
At a minimum, the obtained results should be compared with similar studies for other systems and conclusions should be drawn, for example, on the effect of the chemical composition of the binder on the activation energy and other parameters.
Another question: why was DSC carried out in nitrogen and not in air?
Reviewer 2 Report
The authors report the curing kinetics of fiber-glass reinforced epoxy resin prepregs and their bending properties through DSC and bending tests. While the experimental motive, methods, and results are sound some minor issues must be addressed prior to publication. Comments are detailed below:
- Why was only three point bending tests of the mechanical properties measured? The authors should investigate at least a secondary measure of mechanical properties such as impact strength or modulus.
- Chemical structures of epoxy resins should be described. Were they aliphatic or cycloaliphatic resins? What were the hardeners and curing agents? This is important information that must be included for readers to understand the curing kinetics as aliphatic and cycloaliphatic epoxies have differing curing rates.
- How were the authors able to determine curing degree other than DSC? The authors should include FTIR data to ensure complete consumption of epoxy groups in the supplementary information or within the manuscript. A good example of such is reported by Lee, et. al. https://pubs.rsc.org/en/content/articlehtml/2014/ra/c4ra08289c
Round 2
Reviewer 1 Report
Can be published in current version
Reviewer 2 Report
The manuscript is now acceptable for publication.